# Impact of COVID-19 on Physical Activity in Families Managing ADHD and the Cyclical Effect on Worsening Mental Health

**DOI:** 10.3390/brainsci13060887

**Published:** 2023-05-31

**Authors:** Erica Seal, Julie Vu, Alexis Winfield, Barbara Fenesi

**Affiliations:** 1Faculty of Education, Western University, London, ON N6G 1G7, Canada; eseal@uwo.ca (E.S.); awinfie3@uwo.ca (A.W.); 2Department of Psychology, Faculty of Social Sciences, Western University, London, ON N6A 5C2, Canada; jvu32@uwo.ca

**Keywords:** attention deficit hyperactivity disorder (ADHD), physical activity, COVID-19, barriers to physical activity

## Abstract

Physical activity supports symptom management in children with ADHD and reduces the mental health burden associated with caregiving for children with ADHD. Survey-based research shows that COVID-19 reduced physical activity among diverse populations. This study used a qualitative approach situated within a socioecological framework to (1) understand how COVID-19 impacted physical activity of children with ADHD and their caregivers, to (2) identify barriers to their physical activity, and to (3) identify potential areas of support. Thirty-three participants were interviewed between October 2020 and January 2021. Content analysis revealed that physical activity declined for children and caregivers; significant barriers were social isolation and rising intrapersonal difficulties such as diminishing self-efficacy and energy levels and increased mental health difficulties. Worsening mental health further alienated caregivers and children from physical activity, undermining its protective effects on ADHD symptom management and mental wellbeing. Participants identified needing community support programs that offer virtual, live physical activity classes as well as psycho-emotional support groups. There is vital need to support physical activity opportunities during high-stress situations in families managing ADHD to buffer against diminishing mental wellbeing. This will promote further physical activity engagement and allow families to reap the cognitive, psychological, and emotional benefits.

## 1. Introduction

Attention deficit hyperactivity disorder (ADHD) is one of the most common childhood disorders [1,2]. An estimated 6.4 million school-aged children (11%) have a lifetime diagnosis worldwide [3]. ADHD is characterized by developmentally excessive levels of inattention, hyperactivity, and impulsivity that interfere with daily functioning [2,4]. Eighty percent of children with ADHD also have co-occurring mood disorders, such as anxiety and depression [5], and often struggle with interpersonal relationships and academic learning [6,7]. A common misconception surrounding children with ADHD is that they are highly physically active, given the hyperactive tendencies of many [6,8]. In reality, children with ADHD are less physically fit and active than their neurotypical peers, engage in more sedentary behaviours, and are twice as likely to engage in other unhealthy behaviours [6,9,10,11,12]. During the COVID-19 pandemic, children with ADHD experienced a further decline in their physical activity participation [13,14,15,16], as public health restrictions limited access to school-based activities, extracurriculars, and outdoor play [17,18]. While most children experienced a decline in physical activity during the pandemic [19,20,21], children with ADHD were already less active than their peers and thus experienced further disadvantage.

Participation in regular physical activity is essential for all children’s physical, psychological, social, and cognitive development [22,23,24,25], with added benefits observed for children with ADHD [26,27,28]. In general, physical activity in childhood is associated with healthier body composition [29], lower blood pressure [30], better physical fitness [31], improved bone strength [32], cardiometabolic health [33], and motor skill development [31,34]. Psychosocially, physical activity in childhood is associated with improved mental wellbeing [35], social skills [36], increased self-esteem [36], quality of life [29], resilience [37], and less psychological distress [31]. Among children with ADHD, engaging in physical activity is associated with better symptom management in both cognitive and psycho-emotional domains [27,38,39]. Children with ADHD have functional and structural differences in neural anatomy; functionally, they are observed to have frontal and cingulate hypoactivation, and structurally they present with differences in corpus callosum, cerebellum, and basal nuclei structures [40]. The frontal cortex, and especially the prefrontal regions, are essential in higher-order cognition such as distraction inhibition, sustaining attention, and working memory [41]. Critically, previous research has shown that engaging in regular physical activity may help modify and regulate the structure and functions of the brain that underlie cognition and behavior, as well as the underlying physiology present in ADHD [12,42]. Furthermore, a recent meta-analysis found that physical activity was a major contributing factor to improving anxiety, depression, aggressive behaviours, and social problems often experienced in ADHD [43]. In contrast, inadequate physical activity among children with ADHD is associated with poorer executive functioning (e.g., attention, working memory, inhibitory control), increased symptoms of anxiety and depression, emotional dysregulation, defiant behaviour, and reduced motivation to learn [6,16,44,45,46]. During the pandemic, children with ADHD who engaged in more physical activity and less screen time had fewer externalizing symptoms (e.g., inattention, hyperactivity, oppositionality) as well as fewer internalizing symptoms (e.g., anxiety, depression). This is especially important as research indicates that both externalizing and internalizing behaviours were significantly greater during the pandemic among children with ADHD [47], impairing their quality of life and hampering caregiver–child interactions.

Caregivers of children with ADHD also benefit from physical activity. The responsibilities associated with caring for children with developmental disorders often provoke increased stress, anxiety, depression, and poorer quality of life [28,48,49]. However, caregivers who engage in more physical activity are better able to manage the psycho-emotional challenges of caregiving. Physical activity interventions have been shown to reduce stress, depression, and burden in caregivers [50,51]. Unfortunately, recent work has demonstrated that the physical activity levels of caregivers were also negatively impacted during the pandemic [52]. Caregivers of children with ADHD had to adopt additional roles of teacher and fulltime homekeeper, on top of existing roles and stressors such as a lack of access to therapeutic resources, greater financial hardship and uncertainty compared to other families, and compounding mental health issues [53,54]. It is unsurprising that caregivers could not participate in regular physical activity under such high-stress circumstances, moving these vulnerable individuals even further away from the benefits of physical activity and other self-care behaviours.

The dynamic developmental theory of ADHD emphasizes that ADHD symptomology reflects an interplay between individual predispositions and environmental influences [55]. The particular expression of ADHD symptoms at any given time in life will vary depending on the environmental influences at that time. Thus, during high-stress situations, ADHD symptoms tend to worsen, creating a paradox whereby it becomes increasingly more difficulty to engage in health-promoting behaviours such as physical activity but also progressively more important to engage in those behaviours to reap the cognitive, psychological, and mental health benefits that help manage ADHD symptoms. Intervening with health-promoting behaviours during times of high stress will also support caregivers and the family system, as worsening ADHD symptoms are linked to deteriorating parent–child interactions and further decline in familial wellbeing.

Importantly, families function as symbiotic ecosystems, with caregivers’ physical activity directly influencing their children’s physical activity [56]. In other words, more active caregivers yield more active children. Indeed, every additional 20 min of physical activity completed by a caregiver has been shown to produce an additional five minutes of daily physical activity among their child(ren) [57]. Caregivers of children with ADHD who are physically active at least three hours per week are more than four times as likely to have children who are physically active compared to caregivers who are less physically active [58]. The influence of caregiver physical activity perception and behaviour is even more prominent when children are younger, given their reliance on caregiver financial and transportation support to engage in many kinds of physical activity. Indeed, given how interrelated children and caregivers are when it comes to physical activity participation, it is important to capture how the pandemic has impacted both caregivers and their children.

The socioecological model (SEM) [59,60,61] supports examining interrelated and dynamic factors to accurately reflect determinants of physical activity behaviour. Specifically, the SEM recognizes that intrapersonal, interpersonal, organizational, community, and policy-level factors all play a role in physical activity participation [59,60,61]. It is imperative to understand how children, caregivers, and their broader environment were impacted during the pandemic to accurately appreciate their physical activity behaviours. Furthermore, while previous research has characterized changes in physical activity participation during the pandemic, minimal work has directly identified the barriers to and supports for physical activity participation among families with children who have ADHD. In alignment with the SEM, families with children who have ADHD are a distinct ecosystem with unique challenges, thus requiring a targeted approach to accurately reflect specific barriers and potential supports. Thus, the current study aimed to answer three research questions: (1) How did the COVID-19 pandemic impact physical activity participation among families with children who have ADHD? (2) What were the barriers to physical activity participation during the COVID-19 pandemic among families with children who have ADHD? (3) What supports do families with children who have ADHD need to better engage in physical activity during a pandemic? The study used semi-structured interviews to provide novel qualitative perspectives directly from caregivers and children who have ADHD [62,63].

### Hypotheses

Based on previous research, it was hypothesized that the most salient effect of the pandemic on physical activity participation among families with children who have ADHD would be a decline in engagement [14,15,16]. Furthermore, it was hypothesized that social distancing mandates [64] and declining mental wellbeing [65] would be significant barriers to physical activity participation for both caregivers and their children with ADHD. Additionally, although identifying the supports needed by families was exploratory in nature, based on previous research with neurotypical children [66] and older adults [67] it was hypothesized that greater access to socially distanced physical activities would be noted as helpful for both caregivers and their children.

## 2. Materials and Methods

### 2.1. Participants

Caregivers of children with ADHD were contacted based on previous affiliation with lab research. Recruitment occurred between October 2020 and January 2021. The institution’s Cognitive Neuroscience Research Registry was also utilized to recruit participants. Snowball sampling was also used by asking participants if they knew of any other families with a child with ADHD who would be interested in participating.

A total of 33 participants from Ontario, Canada, took part in the study, with 15 independent family units. A total of 25 family units were contacted, for a response rate of 60%. Thematic saturation was used to decide when to discontinue data collection [68]. Saturation was met when no additional new information, as pertaining to the specific research questions, was being raised during participant interviews. Consultation on this decision was performed throughout the data collection process among researchers. To be eligible for the study, caregiver participants must have had at least one child with an ADHD diagnosis who was living with them for at least some of the time during the COVID-19 pandemic. Demographic information can be found in Table 1.

### 2.2. Procedure

Prior to engaging in the interviews, participants (caregivers and their children) completed consent and assent forms via email, and caregivers completed an online demographics survey through university-approved software. Following this, participants engaged in a semistructured interview with a researcher via Zoom. The interviews lasted approximately one hour for the caregiver(s) interview and 30 min for the child interview. Both caregiver and child participants were compensated for their time in the form of a CAD 10 Amazon gift card.

### 2.3. Materials

#### 2.3.1. Online Survey

Caregiver participants completed an online demographics questionnaire through Qualtrics XM, an institutionally approved survey tool. The questionnaire included items about age, gender, income, race, and education. Specific questions were also asked about the child participant’s ADHD diagnosis (see Table 1).

#### 2.3.2. Semistructured Interviews

The socioecological model (SEM) was used to create the interview questions. As noted previously, the SEM is often used to identify barriers to behaviour by highlighting the relation between intrapersonal, interpersonal, institutional, community, and policy factors influencing behaviour [69]. The term “barriers” was used throughout the interview, with specific questions aimed at the interrelated levels (e.g., references to intrapersonal, interpersonal, and community factors impacting physical activity for both children and caregivers).

The caregiver interview comprised two parts. The first part asked caregivers about the barriers for their children during the pandemic due to their ADHD diagnosis, and whether there had been changes in the presenting symptoms associated with their child’s ADHD. The second part included questions related to physical activity; the questions probed about the caregivers’ physical activity and their child’s physical activity before and during the pandemic and sought to gain insight into the barriers they experienced to maintaining physical activity engagement. The full interview guide is provided as Appendix A.

The child interview also comprised two parts. The first part asked for demographic information of age and gender. The second part asked two questions: (1) What type of physical activity did you do before the COVID-19 pandemic? How often? (2) Have you been able to participate in physical activity during the COVID-19 pandemic? If yes, what activities do you do? If no, do you wish you could do more? Follow-up questions (prompts) were used if children did not understand the initial questions, including “What has helped you keep doing those activities?” and “What has gotten in the way of you participating in physical activity?”. Children were interviewed separately from their caregivers to offer them full freedom of expression. Children and caregivers were asked age-appropriate questions that were different, but related, to account for possible differences in communication or self-reflection ability.

### 2.4. Qualitative Data Analysis

A post-positivist paradigm was used in data collection and analysis [70]. Given that the current study aimed to understand experiences related to the COVID-19 pandemic, which the researchers were also experiencing, a post-positive approach was most appropriate to account for researcher bias and potential influences of personal experience and background knowledge related to the pandemic [71]. Inductive content analysis was used to analyze interview responses [72]. Content analysis compresses texts into content categories based on explicit codes and was used to examine trends and patterns in transcribed audio-recorded interviews [73,74]. The technique was inductive as content categories were derived from the data. The content analysis aimed to identify, analyze, and report common themes from both parent and child interview transcripts. Specifically, the analysis aimed to identify the effects that the pandemic had on the physical activity participation of families with children with ADHD, the barriers they faced to engaging in physical activity, and the supports they would need to better engage in physical activity during a pandemic.

In the first step of data analysis, digital recordings of interviews were transcribed verbatim using Trint, a professional transcription service. The transcribed interviews were then read over independently by two researchers to help them become familiar with the data. Following a familiarization phase, the two researchers collaboratively generated a preliminary codebook to categorize interview content into meaningful groups based on emerging themes. The preliminary codebook was then applied to all transcripts. Following this, researchers consulted on the success of the codebook in capturing key themes in the data. Any coding discrepancies were reviewed and discussed, and final coding decisions were made. A final codebook was generated and reapplied to all transcripts. All transcriptions were then inputted into MAXQDA (V23), a qualitative computer software program, and the finalized coding protocols were applied. All themes and subthemes are reported in tables below. Excerpts from participants are provided to illustrate key findings.

#### Trustworthiness

Multiple measures were implemented to ensure trustworthiness by considering credibility, dependability, and transparency. To ensure credibility, investigator triangulation was used. Multiple investigators took part in the research process, particularly in the data analysis stage, with investigators working together throughout data analysis to ensure consistency in coding. This contributed to credibility by confirming findings across multiple investigators and by minimizing any research bias [75,76]. Dependability was promoted by creating explicit and repeatable methods through the use of recruitment scripts and interview guides. Detailed methodology and research processes were recorded throughout the research process. The institution’s Research Ethics Board (REB) approved the study (protocol #116190).

## 3. Results

The results are divided into three sections, each representing its corresponding research question. All three research questions are represented as a topic theme, with subthemes generated from data analysis. Themes and subthemes are presented in the order of highest frequency of occurrence, which is defined by the number of times the theme was cited by the participants in their interviews. Representative quotations from the interviews are provided.

### 3.1. Research Question 1: How Did the COVID-19 Pandemic Impact Physical Activity Participation among Families with Children Who Have ADHD?

Table 2 provides a summary of the ways in which the COVID-19 pandemic affected the physical activity of families with children who have ADHD. The most salient theme was a reduction in physical activity, with social distancing restrictions and decreased motivation as the major contributing factors. Interestingly, some families noted an increase in physical activity, with an increase in leisurely physical activity opportunities and at-home workouts.

#### 3.1.1. Theme 1: Decrease in Physical Activity

Both caregivers and children expressed that there was a significant reduction in their physical activity participation during the COVID-19 pandemic. The major contributing factors to these reductions (subthemes) were social distancing requirements, decreased motivation, facility closures, and an overall decrease in physical and mental health.

##### Subtheme 1a: Social Distancing Mandate

Caregivers expressed that the social distancing requirements shut down their children’s extracurricular activities that would have otherwise kept them physically active. Children also expressed school closures as interfering with their ability to play at recess, attend physical education class, and play school sports. Several caregivers noted: “Prior to the pandemic, he was doing gymnastics, and swimming and he was signed up for soccer. And then that dropped to zero. And physical activity is huge for him. For focus and getting outside.” (Participant 11, caregiver); “She [could no longer participate] in horseback riding and dance. She missed all her recitals.” (Participant 3, caregiver); “[They] couldn’t go to a playground, or even go outside sometimes. Their hockey, all their programs, had stopped.” (Participant 5, caregiver). Another caregiver described their loss of access to gym facilities and its impact, “I miss my gym. I used to go three to four times a week because I had a personal trainer. Now I have nothing.” (Participant 5, caregiver). One child also expressed:

“[Before the pandemic], I liked running around, walking, just going outside with my friends at recess, and biking. But now, because of COVID, I’ve kind of shut myself down. I spend less time outside and I’ve been doing a bunch of art, playing video games, and just staying inside”.(Participant 2, child)

##### Subtheme 1b: Decreased Motivation

Caregivers described how they themselves and their children were experiencing a significant decline in their motivation to be physically active. The following excerpts illustrate the decline in motivation: “It’s been really hard to personally engage in physical activity and really hard to get my child to engage as well.” (Participant 11, caregiver); “I thought for so long, just do the exercises in the house, but I just can’t find the motivation.” (Participant 14, caregiver); “Unless the exercise was fun, they did not want to participate. It was a lot of ‘Mom, why would I do that?’” They didn’t want to just exercise for the sake of it.” (Participant 8, caregiver). Another caregiver noted:

“[My children] used to enjoy going for hikes in the woods. I could not pay them to leave the house [now]. Personally, I thought for so long, just do the exercises in the house, but no, I just don’t find the motivation to do that”.(Participant 14, caregiver)

##### Subtheme 1c: Decrease in Overall Health

Caregivers expressed that their children’s overall physical health was negatively affected by the pandemic, with many referencing an increase in poor sleep habits due to an increase in sedentary behaviour during the day. Several caregivers also indicated an increase in poor eating habits, prolonged screentime use, and weight gain. “[My children] are getting less sleep as a result of being more sedentary and burning off less energy. Their nighttime routines have been challenging.” (Participant 6, caregiver); “Screentime is an everyday fight…he has also gained a lot of weight.” (Participant 8, caregiver).

#### 3.1.2. Theme 2: Increase in Physical Activity

Some caregivers and children expressed a greater ability to participate in physical activity. Participants noted that they had more opportunities to engage in leisurely physical activity (outdoor walks, biking, hiking) and they were able to leverage the stay-at-home orders and engage in at-home workouts.

##### Subtheme 2a: Increase in Leisurely Physical Activity

Several participants noted that because their daily routines were less structured, it allowed them to engage in physical activity more casually and conveniently. In addition, because there were few public places to atttend, many families took advantage of walking as a form of physical activity and were able to engage in it more regularly. Several caregivers noted: “We went out with our masks for a daily walk. Twice a day, three times a day, just to get out of the house. That was positive.” (Participant 3, caregiver).

“We had a lot more fun because there were less pressures, which probably allowed my children to be themselves more. Everything takes them so much longer [to do] because they [struggle to] focus and they were often unable to move and [be active]. The pandemic gave them more downtime and time to move around”.(Participant 8, caregiver)

##### Subtheme 2b: Increase in at-Home Workouts

Caregivers specifically expressed finding at-home workouts using online videos or self-directed exercises with basic equipment or body weight only, and that the stay-at-home orders forced them to engage in more activity at home given the limited external options. Several noted: “I’ve started to try to work out from home.” (Participant 10, caregiver); “I used to work out three times a week, but now I’m up to five because I [can do it at home] and I just need it”. (Participant 12, caregiver).

### 3.2. Research Question 2: What Were the Barriers to Physical Activity Participation during the COVID-19 Pandemic among Families with Children Who Have ADHD?

Table 3 provides a summary of the barriers to physical activity during the COVID-19 pandemic among families with children who have ADHD. The most significant barriers to physical activity participation noted by both caregivers and children were increased social isolation, intrapersonal difficulties, and screen time use. Caregivers also noted a decrease in available time to participate in physical activity due to increased caregiver responsibilities and a decrease in respite. Caregivers also mentioned that their child(ren)’s routines were severely disrupted (e.g., sleep, school, therapy, friends), leaving little to no room to consider physical activity.

#### 3.2.1. Theme 1: Increased Social Isolation

Caregivers expressed that social isolation was the most significant barrier to physical activity participation for their children. Many caregivers noted that their children’s primary way of being physically active prior to the pandemic was through peer play and socializing, and that social distancing limited these options: “A huge challenge and barrier for [my son] has been the lack of peer play, and a lack of being able to socialize.” (Participant 10, caregiver); “For my son, [physical activity] only happens if it’s social. He’s not a kid who’s going to go out and run around by himself.” (Participant 2, caregiver).

#### 3.2.2. Theme 2: Increased Intrapersonal Difficulties

Caregivers expressed an increase in intrapersonal difficulties both within themselves and within their children that interfered with physical activity participation. Most significantly, caregivers noted a decrease in self-efficacy with new ways of being active, increased mental health difficulties that overrode physical activity priorities, and increased fatigue and lethargy.

##### Subtheme 2a: Decreased Self-Efficacy

Many caregivers described a reduction in self-belief that they could participate in new ways of being physically active, and that their children were shellshocked and unsure how to navigate such a novel routine and a less structured environment. Several caregivers noted: “I was great at doing structured activities [with my child], but I’m not good at playing. I’m a very poor playmate”. (Participant 11, caregiver); “[My child] just doesn’t know how to go about being physically active in this new way.” (Participant 6, caregiver).

##### Subtheme 2b: Increased Mental Health Difficulties

Caregivers expressed their own mental health struggles, and the relation to diminished physical activity and health-promoting behaviours. They also described how their children were struggling emotionally and the consequences for how they engaged with physical activity. One caregiver noted, “I’m not active and I’m making poor food choices because I don’t feel great”. (Participant 11, caregiver). Another caregiver described:

“You can tell that [my children] are hurting by the way they [speak and act]. They just aren’t able to do what they used to, and their independence has been stripped. [My son] went for a bike ride two weeks ago and he froze [it was so cold]”.(Participant 13, caregiver)

##### Subtheme 2c: Decreased Energy Levels

Caregivers noted that the increase in mental health issues due to a variety of factors (e.g., increased caregiver roles, greater uncertainty, and fear) depleted both their own and their children’s energy to engage in physical activity. Many noted that physical fatigue was not the reason for their low energy, but it was the mental fatigue that prevented them from being able to select health-promoting behaviours. Some caregivers expressed: “There is very little energy in the house right now invested in physical activity.” (Participant 14, caregiver); “My son and I are way less energetic, way more sedentary than we used to be. And that’s despite knowing how much physical activity helps him.” (Participant 11, caregiver). Another caregiver mentioned:

“At work, I used to do things like take the stairs instead of the elevator. But that’s disappeared because I’m just so tired. It’s not even physically tired, I’m mentally tired. I take the elevator now because [the pandemic] has made making those small choices really hard”.(Participant 11, caregiver)

#### 3.2.3. Theme 3: Increased Screen Time

Caregivers described how the use of screen time not only increased but also often substituted for time that could have been spent participating in physical activity. Caregivers also emphasized that screen time was often a saving grace, allowing them to focus on other tasks while their children were occupied. However, they recognized the detrimental impact of increased screen time on other areas of their child’s life, such as less physical activity involvement. Several caregivers articulated: “My son’s screen time has doubled.” (Participant 12, caregiver); “As a parent you’re trying to manage everything, so you often default to screen time in order to get other stuff done, [and then they miss out on other things]”. (Participant 10, caregiver); “They’re sitting around all day watching TV or playing video games, or maybe they build a Lego, but they’re not as active as they used to be”. (Participant 5, caregiver). A child participant expressed:

“There’s not much left for me to do and there’s not much that I want to do. I’m trying to take my mind off things [with video games]. It’s also one of the ways I’m trying to interact with people”.(Participant 2, child)

#### 3.2.4. Theme 4: Decrease in Available Time

Caregivers expressed that a significant barrier to physical activity participation during the pandemic was simply less available time. The major contributors to diminished available time were noted as an increase in caregiver responsibilities and an overall decrease in respite.

##### Subtheme 4a: Increase in Caregiver Responsibilities

Caregivers described being overwhelmed with additional caregiving responsibilities during the pandemic, with many becoming stay-at-home caregivers managing their children’s educational, physical, and emotional needs, while also being fulltime employees. Many caregivers described not having “anything left” and could not devote physical energy to another task: “I just can’t handle any more [responsibilities]. I’m so overwhelmed with my own emotions and work, and I’m just trying to get through the day.” (Participant 9, caregiver). Another caregiver described:

“My [physical activity routine] went out the window because I was on practicum Monday, Wednesday, Friday, and I was in school from 8:30 a.m.–5:30 p.m. on Tuesdays and Thursdays, and then preparing dinner for my children and managing the rest of their schedule. I didn’t have time to [be physically active] during the pandemic”.(Participant 10, caregiver)

##### Subtheme 4b: Decrease in Respite

Many caregivers expressed that they were unable to take time away from their children or their household responsibilities to be physically active. Limited options with daycares, babysitters, and an overall diminished social network due to social distancing mandates prevented opportunities for respite and self-care. Caregivers noted: “There has been nowhere to put my kids, and nobody to watch them for me to be able to go to workout or go to the gym.” (Participant 10, caregiver); “Even things like hiring a babysitter has become so complicated. You don’t know who is in their bubble. So the lack of child care has made it really challenging to be active.” (Participant 11, caregiver).

#### 3.2.5. Theme 5: Dysregulated Routine

Caregivers and children expressed that the upheaval in their typical routines was a source of stress. Children discussed how their routines were a source of comfort and caregivers described how disordered routines were major contributors to poor focus, poor sleep, mood disturbances, and overall decreased quality of life for themselves and their children with ADHD. Caregivers articulated: “Before COVID, [my son] knew what was happening. As long as he had structure, for the most part, things weren’t a struggle for him”. (Participant 5, caregiver); “My daughter needs a very specific routine, a structured environment. With COVID there was so much unreliability, it made it really difficult.” (Participant 3, caregiver). A child participant expressed:

“[The hardest thing about dealing with the pandemic] is not having a schedule. I’m used to having to wake up, and within a 10-min period having to get ready, catch a bus, and school was also all the same thing. I want to get my old schedule back, so that I’m not doing random things each day”.(Participant 4, child)

### 3.3. Research Question 3: What Supports do Families with Children Who Have ADHD Need to Better Engage in Physical Activity during a Pandemic?

Table 4 provides a summary of the supports identified by families with children who have ADHD to better engage in physical activity during a pandemic. Caregivers indicated that community supports in the form of physical activity programming and psycho-emotional support groups would help them become more involved in physical activity. Many also expressed a sense of hopelessness and were unable to suggest potential supports as they articulated that the circumstances felt too overwhelming.

#### 3.3.1. Theme 1: Community Supports

Caregivers expressed that they needed more support from community groups that could advocate and information-share on behalf of families raising children with ADHD to improve accessibility to helpful resources. They also emphasized the need for targeted physical activity supports for their children to participate in physical activity more easily and effectively, as well as community organizations to support the psycho-emotional wellbeing of themselves and their children. Some caregivers emphasized: “If there was an advocacy group that could advocate for parents [with children who have ADHD]… otherwise unless your doctor refers you to somewhere, you don’t know what resources are available to you”. (Participant 9, caregiver); “We need more direct, one-on-one support.” (Participant 3, caregiver).

##### Subtheme 1a: Physical Activity Community Supports

Caregivers emphasized that they would appreciate learning skills and techniques to support their children’s physical activity participation at home, rather than needing to leave the home under pandemic conditions. They also expressed that online physical activity options were only effective if it involved a live person leading activities, otherwise their children would disengage from prerecorded videos. Caregivers mentioned: “We would benefit from being taught strategies that we could implement at home…so that we could meet their needs at home.” (Participant 8, caregiver); and

“Businesses and community centers need to think about how to bring physical activity into people’s houses in a virtual format. Simply playing a YouTube video and asking my kids to dance along was not enough. My kids looked at me like I was nuts. If it’s novel and there’s a person live on screen, not just a recording of someone saying do this, do that, my kids would be much more likely to participate”.(Participant 10, caregiver)

##### Subtheme 1b: Psycho-Emotional Community Supports

Caregivers discussed wanting community programming that offered a space to share challenges and resources with families also raising children with ADHD. They described how addressing the mental burden of the pandemic on the lives of both caregivers and children could potentially create greater mental space to then direct towards engaging in physical activity. Some mentioned: “I really wish there was a group of similarly struggling people that could come together and talk about their experiences and they how cope.” (Participant 6, caregiver); “I wish there were groups for children with ADHD, so that they could go hear other kids with similar experiences and the challenges that they’re having.” (Participant 8, caregiver).

#### 3.3.2. Theme 2: Unsure of Supports Needed

Many caregivers were unsure what supports they would need to better engage in physical activity during the pandemic. Many were unable to consider how their daily tasks could shift or be supported to create space for more physical activity. In other words, there was a sense of unrelenting overwhelm that was impervious to support. One participant stated directly, “I don’t know what I need”. (Participant 12, caregiver). A smaller number of participants expressed being sufficiently supported.

##### Subtheme 2a: Hopelessness

Many caregivers described feeling hopeless in their current situation, and that there were no supports that could really help. Rather, the pandemic itself needed to shift for changes in their daily life to occur, which could offer more space for physical activity engagement. Caregivers described: “Unless your doctor refers you to somewhere you don’t know what resources are available to you. We feel helpless”. (Participant 9, caregiver); “I can’t think of anything that can help us be better supported. Trouble is, when everyone’s [stuck] inside doing nothing because you can’t do anything, there’s not much you can do.” (Participant 5, caregiver); and “I don’t know if there’s really anything that can help in this situation…you kind of just have to do with what you have”. (Participant 15, caregiver).

##### Subtheme 2b: Sufficiently Supported

Some caregivers expressed feeling sufficiently supported, and that their inability to engage in physical activity was more a reflection of diminished motivation due to overwhelm in other areas of their lives: “No supports needed. [We are just unmotivated].” (Participant 1, caregiver).

## 4. Discussion

The current study examined how the COVID-19 pandemic affected the physical activity behaviour of families with children who have ADHD. The most frequently identified impact was a reduction in physical activity engagement due to social distancing requirements, decreased motivation, facility closures, and a decrease in overall health. A smaller proportion of families, however, noted an increase in physical activity behaviour, as they were able to engage in more leisurely physical activity throughout the day and many found solace in at-home workouts. The most frequently identified barriers to physical activity participation were social isolation, increased intrapersonal difficulties, increased screen time, decreased available time, and dysregulated routines. Additionally, worsening mental health due to inactivity and circumstance further distanced caregivers and children from physical activity. In turn, this undermined the potential for physical activity to support ADHD symptom management and child and caregiver mental wellbeing. While many families expressed a need for more community supports to facilitate physical activity involvement, several other families expressed feelings of hopelessness and were unable to identify ways they could be better supported. The following will discuss the major themes and subthemes and offer suggestions for future support.

### 4.1. Research Question 1: How Did the COVID-19 Pandemic Impact Physical Activity Participation among Families with Children Who Have ADHD?

#### 4.1.1. Decrease in Physical Activity

Both caregivers and children noted that their physical activity participation declined during the pandemic. This finding is supported by prior work that also found a decrease in physical activity among children with ADHD during the pandemic [14,15,16], as well as among caregivers [52]. Participants attributed the decrease in physical activity to social distancing requirements, decreased motivation, facility closures, and a decrease in overall health.

##### Social Distancing Mandate

Caregivers and children cited that government-mandated social distancing measures were the most prominent contributors to reductions in physical activity during the pandemic. This was especially true for children as they lost the ability to participate in peer play, organized sports, physical education classes, indoor/outdoor playgrounds, and extracurriculars. Many caregivers noted exerting excessive effort to encourage physical activity participation among their children, which was often met with resistance. Children between 7–12 years of age often require heavy involvement from caregivers to engage in physical activity, and this becomes compounded when there is no support from school, extracurriculars, or organized sports. Unfortunately, children were not interested in or motivated to partake in activities organized by their caregivers, as they often found them unengaging and isolating. Instead, they opted for sedentary activities such as watching TV or playing video games, which were more entertaining and allowed them to connect with peers. Recent systematic reviews among youth aged 5–17 years similarly found that social distancing mandates and pandemic-related closures were the major contributors to hindered physical activity participation [14,77]. Other studies also suggest that lack of adequate information surrounding COVID-19 may have contributed to increased fear and greater desire to socially distance [78,79,80,81].

##### Decreased Motivation

Caregivers shared that their children with ADHD experienced a significant decline in motivation to participate in physical activity. Caregivers expressed that they appreciated the importance of physical activity for wellbeing, but that their children did not share a similar appreciation unless physical activity was offered in an enjoyable way. Many caregivers shared that it was a “battle” to motivate their children to engage in any physical activity, unless the activity was fun, engaging, or purposeful. Children resisted physical activity for the “sake of” physical activity, such as simply going for walks or participating in at-home exercises, despite caregiver encouragement and modeling. Although previous research has shown that direct caregiver support and behavioural modeling are strong predictors of children’s physical activity participation [82], these were not sufficient qualities to promote physical activity among children with ADHD during the pandemic. Children and caregivers both shared that motivation was directed towards sedentary pastimes, especially screentime, displacing time that could be directed towards physical activity. These findings may be related to the documented increases in mood-related disturbances among caregivers and children with ADHD during the pandemic, including amplified depression and anxiety [16,83,84,85,86,87,88]. Problematically, poor mental health is a barrier to engaging in physical activity despite the act of engaging in physical activity helping to reduce mental distress. The decrease in motivation is likely partially due to the impact of pandemic-related stressors on mental wellbeing, which may have then manifested as a lack of desire to engage in physical activity [65]. For children, the lack of enjoyment surrounding the types of physical activity options available was also a major deterrent; this coincides with previous work arguing that experiencing joy during physical activity is critical for voluntary and sustained participation [89,90,91,92].

##### Facility Closures

Several caregivers expressed that the closure of facilities (e.g., gyms, recreation centers) was a significant contributor to their decline in physical activity during the pandemic. Caregivers were emphatic that their prepandemic ability to leave their home and either attend the gym or attend other physical activity events (e.g., yoga, spin class) was imperative to staying active. Given social distancing requirements, many caregivers were left without the resources or knowledge on how to remain active in alternative ways. While previous research has also shown that facility closures contributed to a decline in physical activity participation among the general public [64], the current study elucidates that those facilities not only provided resources but also important social facilitation. Several caregivers indicated that their motivation to engage in physical activity was directly linked to being surrounded by others participating in physical activity [93]. Thus, the lack of environmental support contributed to poor motivation and declining physical activity. Caregivers also expressed that recreation center closures undermined their child(ren)’s physical activity participation, as many were involved in hockey, dance, swimming, and other sports.

##### Decrease in Overall Health

Caregivers spoke about how their children’s overall physical and mental health declined, contributing to diminished physical activity. Specifically, caregivers noted that their children’s sleep quality was negatively affected; they often went to sleep late and were unable to wake in the morning. As a result, children had less energy to engage in physical activity during the day. This resulted in a cyclical negative feedback loop wherein poor sleep routines undermined children’s ability to participate in physical activity, which produced an excess of energy at the end of the day, and further disturbed sleep onset and quality. This is especially problematic among children with ADHD, as they already experience sleep difficulties [94,95], which were further amplified by the pandemic [96]. Some caregivers mentioned that their children experienced an increase in general aches and pains, which they attributed to excessive sedentary behaviours; this further undermined children’s desire to participate in physical activity. Unhealthy weight gain was also mentioned by caregivers as a contributor to their child(ren)’s lack of physical activity participation, which was again attributed to increased sedentary behaviour and screen time. Many caregivers spoke about their own decline in mental health, and how it interfered with their ability to practice self-care behaviours, such as engaging in physical activity. This mirrors previous research that similarly found that the pandemic negatively affected several elements of health among both caregivers and children with ADHD [16,83,84,85,86,87]. It is evident that the decline in physical activity was associated with a decline in other health-promoting behaviours, including sleep, diet, and routine, which are essential lifestyle factors that impact ADHD symptom management.

#### 4.1.2. Increase in Physical Activity

Interestingly, some caregivers and children expressed that they were able to participate in more physical activity during the pandemic, as they were able to engage in more leisurely and at-home activities. Participants noted that the lack of daily routine and structure had some positive benefits by allowing more opportunities to go outside for walks or bike rides at random, unscheduled times; previously, physical activity was often scheduled in the form of school, gym, or extracurriculars. Furthermore, a few caregivers noted that since at-home workouts became the only option at times, the accessibility of online videos was motivating and helped improve their engagement in physical activity compared to prepandemic [97]. These results are similar to work showing that the pandemic disproportionately negatively affected the activity levels of youth, but not adults (18 years and older). In Canada, recent work found no change in physical activity participation among those between 18 and 49 years of age during the pandemic, while those 50 years of age and older experienced an increase in physical activity [98]. Other work has also shown that among university students in parts of Spain, weekly physical activity participation also increased during the pandemic [99]. More nuanced research among university students also showed that students who were low to moderately physically active participated in more physical activity during the pandemic, whereas those who were highly active participated in less activity [100].

### 4.2. Research Question 2: What Were the Barriers to Physical Activity Participation during the COVID-19 Pandemic among Families with Children Who Have ADHD?

#### 4.2.1. Increased Social Isolation

The most frequently expressed barrier to physical activity for both caregivers and children with ADHD was social isolation. Participants spoke at length about the difficulties of implementing physical activity into their routine when the social aspects were unavailable due to stay-at-home orders or social distancing mandates. This was especially salient for children with ADHD, as many caregivers discussed the resistance and disinterest their children had for physical activity when it no longer involved peers. Indeed, peer play and extracurriculars were the primary ways that these children engaged in physical activity prior to the pandemic. Social isolation not only removed these opportunities for activity, but it also created a significant gap in children’s days that otherwise would have been spent playing with peers or participating in extracurricular activities. The SEM would characterize the decline in physical activity due to increased social isolation as reflecting fractures to all levels of the system, including the intrapersonal level (e.g., negative attitude towards caregiver-led activities), the interpersonal level (e.g., lack of engagement with peers and coaches), the institutional level (e.g., lack of school-based physical activity opportunities, lack of organized sports), the community level (e.g., lack of community-organized activities), and the policy level (e.g., social distancing mandates preventing interpersonal interactions which were the bedrock of prepandemic physical activity participation). Previous research similarly found that social isolation was one of the most significant barriers to physical activity participation among diverse populations [64]. Although social isolation was problematic for many families, those raising children with ADHD rely even more so on social elements to provide routine, structure, and predictability in daily life [54].

#### 4.2.2. Increased Intrapersonal Difficulties

Participants frequently shared that increased intrapersonal difficulties were a significant barrier to participating in physical activity during the pandemic. Many internal factors—biological, personal, mental—interfered with caregivers’ and children’s ability to participate in physical activity. Participants expressed that intrapersonal factors related to decreased self-efficacy (i.e., belief in your ability to accomplish a task), decreased energy levels, and increased mental health difficulties interfered with physical activity involvement.

##### Decreased Self-Efficacy

Several caregivers expressed a hopelessness in their ability to motivate either themselves or their children to be physically active. As previously mentioned, these children were highly disinterested in the available options for physical activity (e.g., walking, hiking, video-based activities) and were unmotivated when it did not involve peers. Caregivers discussed how creating engaging, novel, and purposeful physical activity for their children required immense creativity and effort. This, in turn, left caregivers feeling burnt out and at a loss for new ideas, further diminishing their self-efficacy. Some caregivers expressed a similar kind of hopelessness among their children; these caregivers shared that their children did not know how to be active during the pandemic, appearing lost and confused when trying to find ways to actively play without peers. Self-efficacy is a crucial component to the ongoing engagement in any behaviour, as noted by social cognitive theory [101,102]. Importantly, self-efficacy only develops when tasks are successfully completed, and is often supported by a knowledgeable teacher or peer. Given the lack of access to knowledgeable models and peers beyond the caregivers, the decline in self-efficacy is not surprising. Perhaps older children (12+) could have been more capable of developing self-efficacy outside of caregiver influence, but child participants in the current study clearly struggled, with only their caregivers to support their self-efficacy development. Although a decrease in self-efficacy can be viewed as an intrapersonal barrier, it is a manifestation of misaligned interpersonal connections (e.g., lack of peer interactions, poor communication with caregivers), a lack of institutional support (e.g., school closures, no extracurricular options), a dearth of community support, and a direct reflection of the public health mandate of social distancing which minimized opportunities for physical activity participation.

##### Decreased Energy Levels and Decline in Mental Health

Some caregivers shared that both they and their children experienced a decline in energy levels and mental health during the pandemic. For caregivers, diminishing energy and mood were largely attributed to the overwhelming and expanding roles that they were required to take on during the pandemic. Their main goal was to conserve as much energy as possible to tackle the many tasks of the day. Some caregivers recognized the cyclical nature of worsening mood and physical inactivity and expressed that their inability to participate in physical activity was also contributing to low mood. For children with ADHD, lack of energy and lower mood were attributed to poor sleep, lack of physical activity engagement, boredom, and excessive screen time. Unfortunately, low energy and low mood further contributes to physical inactivity and poor sleep, compounding the negative effects of lethargy and mental distress on lifestyle. This is especially problematic for children with ADHD as they are already more likely than their typically developing peers to be physically inactive, have poorer sleep quality, and experience comorbid mental health issues [6,9,10,11,12,16,54,83,103]. Even prepandemic, lethargy and mental health difficulties have been noted as common elements in the lives of children with ADHD [104,105], interfering with their ability to focus in school and undermining their learning. This research shows that the pandemic led to further declines in energy and mood among children with ADHD, potentially setting them even further behind their typically developing peers in terms of physical activity engagement and possible learning outcomes. Importantly, engaging in physical activity has consistently been shown to help manage mental health disturbances among both children with ADHD and caregivers [27,28,38,39,50,51,52]. It is imperative that there are more opportunities available for vulnerable youth and their families during times of crises to participate in physical activity to protect their mental wellbeing and allow for further engagement in self-care practices.

#### 4.2.3. Increased Screen Time

Caregivers unanimously expressed an increase in screen time for their children. Many children also noted an increase in screen time during the pandemic. Both caregivers and children shared that the increase was associated with a mixture of boredom and lack of options to participate in other activities but was also used as a way for children to connect with their peers. This is consistent with previous research among children with ADHD during the pandemic, showing an increase in 3–4 h of recommended daily screen time [96]. Since peer play, recess, extracurriculars, and organized sports were unavailable, playing online video games was an alternative way to stay connected. Indeed, a recent systematic review examining the effects of video gaming during the pandemic on children’s cognition and behaviour [106] emphasized that context is essential to consider when determining the value or detriment of video gaming. In other words, in times of extreme social isolation, such as the pandemic, video gaming may have offered a healthy means of connection among youth. As previous themes have unpacked, children were uninterested in participating in physical activities that did not involve play, peers, or fun. Thus, while screen time may be viewed as a barrier to physical activity as it consumed time in a child’s day, from their perspective, the lack of viable physical activity options was the true barrier, and screen time was simply the compensatory activity. Many caregivers expressed feelings of guilt surrounding their child’s increase in screen time. However, many also conveyed that screen time was necessary (and welcome) at times to occupy their children’s attention so they could accomplish various work-related or household tasks. Several caregivers associated their child’s increased screen time with increased lethargy and attributed their declining physical activity to this diminishment in energy. Once again, this suggests a cyclical effect in which increased screen time increased lethargy, leading to physical inactivity, which in turn led to the selection of sedentary behaviours such as screen-based activities and further distanced children from the mental health benefits of physical activity.

#### 4.2.4. Decrease in Available Time

##### Increase in Caregiver Responsibilities and Decrease in Respite

Many caregivers expressed that their ever-expanding responsibilities made prioritizing physical activity extremely challenging in an already demanding schedule. Many conveyed feelings of exhaustion due to a heavy workload in multiple areas, and that physical activity was simply unfeasible. Related to the increase in caregiver responsibilities is a corresponding decrease in caregiver respite. Caregivers articulated that even if they wanted to participate in physical activity, unless it involved their children, it was not possible due to a lack of childcare options. Access to daycares, babysitters, and extended family was limited due to social distancing mandates. Thus, any available time to engage in physical activity was occupied by child-rearing. This theme represents significant institutional, community, and policy-level barriers, as caregivers had little to no options for participating in physical activity due to caretaking demands that could not be temporarily sourced out.

#### 4.2.5. Dysregulated Routine

Many caregivers and children with ADHD articulated that the breakdown in their regular schedules and the corresponding lack of predictability introduced tremendous psychological challenges, interfering with their ability to schedule physical activity into daily routine. Extensive research supports how structure and routine is essential for managing many ADHD symptoms [107,108,109]. Indeed, lack of routine was recently identified as the most significant barrier to positive mental health for caregivers and their children with ADHD [54]. In turn, poor mental health further contributes to diminishing physical activity. This pattern has also been found among Canadian adults [65], with waning mental health impairing motivation to engage in physical activity during the pandemic. From a socioecological perspective, dysregulated routine is a barrier borne out of issues stemming from institutional, community, and policy levels. Given the public health mandates of social distancing, schooling and extracurricular opportunities were vastly modified or altogether unavailable, creating significant ruptures in children’s daily routines. Many caregivers also experienced notable changes to their work life, to their home life, and to their daily responsibilities, fracturing their predictable routines as well. As a result, both caregivers and children experienced a decline in their mental wellbeing [54], undermining their ability to direct mental effort towards scheduling physical activity participation, and further removing the protective effects of physical activity on ADHD symptomology and caregiver wellbeing.

### 4.3. Research Question 3: What Supports Do Families with Children Who Have ADHD Need to Better Engage in Physical Activity during a Pandemic?

#### 4.3.1. Community Supports

##### Physical Activity Community Supports

Caregivers expressed wanting support to learn how to engage their children in physical activity at home, as they felt ill-equipped to provide engaging, novel, and regular physical activity options for their children. Some caregivers attempted to play exercise videos that their children could follow along with, but the lack of a live person leading the activities was quickly deemed “boring” and their children disengaged. Some recommendations included community recreation centers providing virtual classes with a live coach or teacher, so that children could feel more connected and engaged during activity. In a study involving children with obesity, providing an online exercise program during the pandemic improved several health-related metrics, including physical fitness, waist circumference, body mass index, waist-to-height ratio, and improved physical activity levels in general [66]. Among older adults, virtual reality (VR)-integrated exercise during the pandemic was shown to promote motor ability, reduce obesity, and prevent falls [67]. Other global communities also provided virtual physical activity options during the pandemic [110], but many community members were unaware of their availability, pointing to further issues with how resources are communicated. Taken together, there is evidence to suggest that providing virtual, live physical activity options can promote greater physical activity participation among diverse populations. Importantly, however, effective communication regarding resource availability is imperative to successful participation.

##### Psycho-Emotional Community Supports

Caregivers expressed that community programming targeted at both caregivers and children with ADHD that provided a space for resource-sharing and open communication could potentially reduce the emotional burden they were all carrying. Previous research has suggested several avenues to improve mental health support for families with children who have ADHD [54]. For children during times of online schooling, teachers could facilitate a supportive environment whereby children could share their experiences via virtual breakout rooms. Community centers and libraries could similarly organize social programming for families and children with exceptionalities to foster connection. For caregivers, prior work also demonstrates the importance of social connection to help buffer against the negative psychological impact of the pandemic [54]. Given the cyclical nature of mental health and physical activity engagement, the goal is to improve the psycho-emotional wellbeing of caregivers and children with ADHD to allow them the mental fortitude to be motivated to engage in physical activity. Promisingly, engaging in physical activity will further promote mental wellbeing and provide additional protection against the negative effects of high-stress conditions.

#### 4.3.2. Unsure of Supports Needed

##### Hopelessness

Some caregivers were unable to identify potential supports that could help improve their engagement with physical activity during a pandemic. Several expressed feeling hopeless in their situation, and simply needing to “get through it” as the only solution. They viewed the stressors imposed by the pandemic as situations to be endured and eventually overcome, and that physical activity would have to be deprioritized until the pandemic was over. Hopelessness is extremely common during times of high stress [111,112] and can fuel mental health disturbances such as depression and anxiety. As previously uncovered, an unfortunate consequence of hopelessness is that it can exacerbate (or even elicit) poor mental health, which further removes individuals from engaging in physical activity behaviours and undermining the potential benefits it can have on reducing mental health issues and states of hopelessness. Extensive work has shown that engaging in physical activity can reduce feelings of hopelessness among suicidal young adults [113], among men [114], among older adults [115], and even among those in prison [116], with researchers arguing that the benefit of physical activity on hopelessness is due to improvements to mental wellbeing [111].

##### Sufficiently Supported

Similar in concept to the previous theme, a few caregivers noted feeling sufficiently supported, and that their lack of physical activity participation reflected the increased demands on their time and energy in other domains of life. One caregiver circled back to the theme of decreased motivation, and how that was the driving force in their lack of physical activity participation. Although at first glance this subtheme appears optimistic, it further highlights how caregivers felt unable to move beyond the constraints imposed upon them by public policy and social distancing mandates.

### 4.4. Physical Activity Recommendations for Children with ADHD and Their Caregivers

The current study underscores several key factors that are important for supporting physical activity among children with ADHD and their caregivers. First, fun and engaging physical activity is essential for children with ADHD. Community centers and schools should provide virtual, live physical activity options with as much interactivity as possible. Second, routine is imperative to help children with ADHD manage their symptomology and to afford parents their own opportunities for physical activity. Schools need to continue to provide daily physical activity options, whether they are virtual or onsite. Community centers could also offer daily, consistent physical activity classes to promote predictability and structure into daily routine. Both caregivers and children would be aware of their daily schedule and when and where certain activities would be taking place, offering much-needed routine. Third, because mental wellbeing is so intimately tied to the desire to engage in physical activity, providing opportunities for children and caregivers to socially interact with peers can lessen the mental burden of high-stress situations and increase the likelihood of pursing physical activity behaviours. Again, this could be organized through community centers, libraries, and schools, helping generate the resilience to engage in physical activity.

### 4.5. Limitations

While this study contributes to our understanding of the effects of the pandemic on the physical activity behaviours of families with children who have ADHD, it is not without its limitations. First, the sample size was limited and mainly included Caucasian families with post-secondary education. Families from diverse racial, ethnic, and socioeconomic backgrounds may have experienced the pandemic’s effect on physical activity differently and may have identified additional barriers to participation. Related to this limitation is the fact that virtual interviews were used to gather data, which precluded participation of families who did not have access to the internet or a computer. Second, due to challenges with recruitment, interviews were conducted during different times of the pandemic. Some families interviewed later in the data collection process may have been able to attend the gym or may have been able to gather with friends and family, improving their mental wellbeing, sense of connection, and desire for physical activity. Relatedly, families interviewed in the winter months often shared that they were participating less in physical activity due to weather conditions more so than pandemic-related restrictions. Third, the themes identified in the study predominantly reflect the views of caregivers as it was challenging for children to discuss their views of the pandemic. The questions asked in caregiver interviews were more detailed, whereas the questions asked in child interviews were simpler and thus unveiled less information. This may have limited the study’s ability to fully reflect how children with ADHD experienced the effect of the pandemic on their physical activity.

### 4.6. Future Directions

Future research should gather neuropsychological assessment data from child participants to better understand where on the ADHD spectrum they were positioned. This could have provided nuance to the study interpretations, such as perhaps that higher-functioning children with ADHD and their caregivers were less negatively impacted by the pandemic, as they may have had better compensatory strategies during times of high stress. Additionally, it would be informative to gather ADHD diagnostic information about caregivers to better understand how families with multigenerational diagnoses can be better supported, given the compounding challenges that are likely present in such environments. Lastly, it is important to expand this research to more diverse ethnic, racial, and socioeconomic backgrounds, given the unique intersectional stressors that impact these marginalized groups.

## 5. Conclusions

This study aimed to answer how the COVID-19 pandemic impacted physical activity participation among families with children who have ADHD, to identify the barriers to physical activity participation among these families, and to elucidate potential supports that could help families engage in physical activity during a pandemic. As anticipated, the most significant consequence of the pandemic was reduced physical activity participation for both caregivers and their children with ADHD. Interestingly, some families noted an increase in their ability to engage in physical activity due to less rigid schedules and their ability to embrace more leisurely activity such as walking and hiking. The most significant barrier to physical activity participation was social isolation due to public health mandates. Other notable barriers included a decrease in available time due to increasing caregiver responsibilities and diminished respite, as well as dysregulated daily routines. Interestingly, several barriers to physical activity participation yielded a double bind. Specifically, physical activity participation was a viable solution to the same barriers preventing caregivers and their children from engaging in physical activity. These included barriers such as increased intrapersonal difficulties (decreased self-efficacy, decreased energy levels, increased mental health difficulties) and increased screen time. These barriers were a manifestation of varying interpersonal, institutional, community, and policy-level factors; problematically, their presence further distanced caregivers and their children from being able to engage in physical activity, which removed the protective effects of physical activity on ADHD symptom management and mental wellbeing. Families suggested that community supports offering online, live, physical activity opportunities would be beneficial for both themselves and, especially, their children, who sorely missed social interaction and peer play during physical activity. They also suggested psycho-emotional support programming be offered to lessen the mental health burden on both them and their children; this could help create the necessary mental and energetic precursors to participate more regularly in physical activity, and thus allow children and caregivers to reap the cognitive, psychological, and emotional benefits of physical activity. In summary, this research helps elucidate how times of crisis impact health-promoting behaviours among families dealing with ADHD, and the corresponding consequences for ADHD symptom management and mental health, with the hope that more effective safeguards become available in times of need.

## Figures and Tables

**Table 1 brainsci-13-00887-t001:** Demographic characteristics.

Demographic Characteristic	Frequency
Gender of caregiver	
Male	2
Female	13
Gender of child	
Male	12
Female	6
Age of child (y) (Mean/SD)	10.16/2.2 (range 7–12)
Race of parent	
Caucasian	14
Black	1
Education	
Some post-secondary	1
Post-secondary	4
University/professional degree	7
Household income	
20,000–30,000	1
80,000–90,000	1
90,000–100,000	2
100,000+	5
Prefer not to say	6
Comorbid neurological diagnosis (child)	3
Comorbid mental disorder diagnosis (child)	3
Comorbid physical/auditory/visual disorder diagnosis	4

**Table 2 brainsci-13-00887-t002:** Frequency summary of main themes and subthemes of how COVID-19 affected physical activity participation among families with children who have ADHD.

Theme	Frequency
Decrease in Physical Activity	91
Social Distancing Mandate	34
Decreased Motivation	33
Facility Closures	8
Decrease in Overall Health	6
Increase in Physical Activity	19
Increase in Leisurely Physical Activity	9
Increase in At-Home Workouts	3

**Table 3 brainsci-13-00887-t003:** Frequency summary of main themes and subthemes of the barriers to physical activity participation during COVID-19 among families with children who have ADHD.

Theme	Frequency
Increased Social Isolation	39
Increased Intrapersonal Difficulties	32
Decreased Self-Efficacy	17
Decreased Energy Levels	6
Increased Mental Health Difficulties	6
Increased Screen Time	28
Decrease in Available Time	14
Increase in Caregiver Responsibilities	6
Decrease in Respite	5
Dysregulated Routine	13

**Table 4 brainsci-13-00887-t004:** Frequency summary of main themes and subthemes of supports identified by families with children who have ADHD to better engage in physical activity during a pandemic.

Theme	Frequency
Community Supports	14
Physical Activity Community Supports	9
Psycho-Emotional Community Supports	5
Unsure of Supports Needed	11
Hopelessness	6
Sufficiently Supported	3

## Data Availability

The data presented in this study are available on request from the corresponding author. The data are not publicly available due to privacy reasons given the interview-based nature of the data.

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
