# Peer review of "Impact of COVID-19 on Physical Activity in Families Managing ADHD and the Cyclical Effect on Worsening Mental Health"

_brainsci, 2023, doi:10.3390/brainsci13060887_

Round 1
Reviewer 1 Report
I very much enjoyed reading this paper. Thank you for the opportunity to review it. The introduction provided a convincing and detailed overview of the literature in this area and provided a solid foundation for the current study. It was well-written and well-referenced. For the method and results, I do have a few queries which I hope the authors would be able to respond to:
- Participants: Overall this study had a good sample size with 15 independent family units but I wondered what drove the original sample size. Was this information power or saturation point for example. It would be good to have a sentence added in about the rationale for this number.
- Interviews: The interview guide for the caregiver interview is helpfully included in the supplementary material and the interview questions for the children are in the text (lines 162-165). From reading the latter it looks as though children were not asked about barriers or support/enablers. If this is the case, it would be interesting to understand why not? I also see that the parents were not present during the child's interview, which makes sense for the reasons outlined but was it possible to link the data afterwards i.e., so you had family pairings? Given the retrospective nature of the questions I wonder if that would help be sure of accurate responses.
- Parent data: I realise this information may not have been collected but were parents/caregivers asked about ADHD diagnosis for themselves?
- A very minor point, and one which I realise not all will agree with, but the use of frequencies on the qualitative data feels a bit like the data is being quantified and I do not think that is ideal. I realise if you don't do this, another reviewer will ask you to so I would not advocate taking it out but perhaps just removing the replication of the frequency from the Theme Subheading titles.
- For theme one all the example quotes seem to relate to the child's physical activity and not the caregivers but the research question covered both. Are there some caregiver examples that could be included? Or perhaps that is the point i.e. caregivers did more because they did home workouts but children did not?
- For theme three I was interested in whether some of the subthemes are barriers or consequences and how this was decided. For example, the increased screen time could be either.
The discussion to the paper is very thorough but it is also very long. The opening paragraph provides a clear summary of the results but I wonder if after that the discussion could be made a little more concise, perhaps by consider themes together rather than all the subsections. This is no reflection on the quality of the discussion but rather the readability. I think there is a small amount of repetition which could be reduced.
The final part of the conclusion relates these findings to a general crisis rather than COVID-19 but it would be good to extrapolate findings more because (we hope) we are now at the end of the pandemic. I wonder if a paragraph could be inserted into the discussion which looks at what factors are clearly important in exercise for those with ADHD e.g. routine, fun etc. Almost as a set of recommendations to keep children and their caregiver's active. I think this will make it easier for other authors to see the work's implications.
Author Response
We sincerely thank you for your thoughtful feedback and suggestions. Our responses are provided in a word document for easier readership. However, they are also pasted below.
Reviewer 1
Comment 1: I very much enjoyed reading this paper. Thank you for the opportunity to review it. The introduction provided a convincing and detailed overview of the literature in this area and provided a solid foundation for the current study. It was well-written and well-referenced. For the method and results, I do have a few queries which I hope the authors would be able to respond to.
Response to Comment 1: Thank you for the thoughtful comments and support, it is much appreciated!
Comment 2: Participants—Overall this study had a good sample size with 15 independent family units, but I wondered what drove the original sample size. Was this information power or saturation point for example. It would be good to have a sentence added in about the rationale for this number.
Response to Comment 2: Thank you for noting this gap in information provided. We used thematic saturation to decide when to discontinue data collection. Saturation was met when no additional new information as pertaining to the specific research questions was being raised during interviews. Consultation on this decision was done throughout the data collection process among researchers. We have included this information on page 4 (line 150-153).
Comment 3: Interviews—The interview guide for the caregiver interview is helpfully included in the supplementary material and the interview questions for the children are in the text (lines 162-165). From reading the latter it looks as though children were not asked about barriers or support/enablers. If this is the case, it would be interesting to understand why not? I also see that the parents were not present during the child's interview, which makes sense for the reasons outlined but was it possible to link the data afterwards i.e., so you had family pairings? Given the retrospective nature of the questions I wonder if that would help be sure of accurate responses.
Response to Comment 3: We appreciate that it was unclear how we asked child participants about barriers/facilitators of physical activity during the pandemic. We have clarified that the questions: “Have you been able to participate in physical activity during the COVID-19 pandemic? If yes, what activities do you do? If no, do you wish you could do more?” were designed to target these exact barrier/facilitator elements with child-friendly language. For some children, these questions were enough to get them to describe barriers/facilitators, but for others it was not. This was the main reason for using semi-structured interviews, which allowed for us to elaborate on the lead questions with clarifying (prompt) questions if necessary, such as “What has helped you keep doing those activities?” and “What has gotten in the way of you participating in physical activity?” We have included descriptions of these prompts on page 4 (lines 191-194). Indeed, children did discuss barriers to their physical activity participation such as increased screen time and dysregulated routine (page 10 line 393 and line 433), suggesting that the interview questions targeted the key constructs we aimed to understand.
Unfortunately, the child and caregiver data were anonymized and stored separately, so their data cannot be linked. However, we completely appreciate how this information would be helpful and will reconsider how child and caregiver data are matched in future work.
Comment 4: Parent data—I realise this information may not have been collected but were parents/caregivers asked about ADHD diagnosis for themselves?
Response to Comment 4: This is an interesting point. Unfortunately, caregivers were not asked about their own personal ADHD status. However, this would be interesting to consider given the intergenerational nature of ADHD and whether overlapping ADHD symptomology within a family dynamic further exacerbates barriers to physical activity during times of high stress such a pandemic. This point has been included on page 20 (section 4.6, line 877) as a promising topic for future research.
Comment 5: A very minor point, and one which I realise not all will agree with, but the use of frequencies on the qualitative data feels a bit like the data is being quantified and I do not think that is ideal. I realise if you don't do this, another reviewer will ask you to so I would not advocate taking it out but perhaps just removing the replication of the frequency from the Theme Subheading titles.
Response to Comment 5: The purpose of adding a quantifiable metric like frequency to the data was to reflect the importance of a given theme in the lived experience of caregivers and their children. In other words, the more frequently mentioned a theme was, the more important it was in these participants’ collective experiences. Thus, we wanted to reflect a hierarchy of relevance to some degree. But we are happy to remove the replication of the frequency from the Theme Subheading titles. We sincerely appreciate the thoughtfulness of this comment and the reviewer’s awareness that other reviewers may indeed ask for more quantifiable data (as another reviewer has done).
Comment 6: For theme one all the example quotes seem to relate to the child's physical activity and not the caregivers, but the research question covered both. Are there some caregiver examples that could be included? Or perhaps that is the point i.e., caregivers did more because they did home workouts but children did not?
Response to Comment 6: Thank you for pointing this out. We have included caregiver quotes that pertain to themselves (not just their children) on page 7 (line 269, line 281).
Comment 7: For theme three I was interested in whether some of the subthemes are barriers or consequences and how this was decided. For example, the increased screen time could be either.
Response to Comment 7: This is a great point. Indeed, several of the subthemes could be viewed in this light e.g., barrier and consequence. It was determined that the themes/subthemes were barriers based on the context within which they were spoken about by participants. For example, caregivers spoke about how increased screentime was a consequence of the pandemic (due to boredom, fatigue, etc.), and how as a result it evolved into a barrier to physical activity. This important nuance is represented on page 17 (line 732-738).
Comment 8: The discussion to the paper is very thorough but it is also very long. The opening paragraph provides a clear summary of the results, but I wonder if after that the discussion could be made a little more concise, perhaps by considering themes together rather than all the subsections. This is no reflection on the quality of the discussion but rather the readability. I think there is a small amount of repetition which could be reduced.
Response to Comment 8: Thank you for this comment. We have tried to minimize repetition and increase conciseness throughout the discussion. In addition, we removed the subtheme “increase in leisurely physical activity and at-home workouts”, condensed it, and combined it with the theme “increase in physical activity”. Furthermore, we consolidated and combined subthemes “decreased energy levels” and “increase in mental health difficulties” into the subtheme “decreased energy levels and decline in mental health”. We hope these changes have sufficiently removed unnecessary repetition.
Comment 9: The final part of the conclusion relates these findings to a general crisis rather than COVID-19, but it would be good to extrapolate findings more because (we hope) we are now at the end of the pandemic. I wonder if a paragraph could be inserted into the discussion which looks at what factors are clearly important in exercise for those with ADHD e.g. routine, fun etc. Almost as a set of recommendations to keep children and their caregiver's active. I think this will make it easier for other authors to see the work's implications.
Response to Comment 9: Thank you for this suggestion. We have included a paragraph emphasizing the key factors involved in physical activity for those with ADHD and their caregivers and included a set of recommendations to keep children and their caregivers’ active (page 19, section 4.4, line 837).

Reviewer 2 Report
Dear authors, I read the paper with interest and I agree with a need of more qualitative research. However, I find a few aspects of the article, which might be improved. There is no kind of theoretical psychological background. There is no references and comments about the age of the children with ADHD. Also, the mention of the age of the children is very important for understanding of organization of their activities. Outdoor activities, the guidance and orientation from the adults changes according to psychological age of the children and adolescents. This aspect should be covered and commented in the article. Also, all the results are based only on the responses of the questions of the interviews. A strong limitation is the absence of any kind of professional psychological or neuropsychological assessment of the population of children with ADHD. Some recent neuropsychological studies (Machinskaya and Cols.,) have shown that children with ADHD frequently show insufficient level of brain subcortical activation (brainstem system), the data, which is close to the introduction of the paper. It is important to include citations from clinical recent studies (Glozman and Cols., 2020).
It is also important to mention, that the families, actually, had opportunity to spend time outside during Pandemic. The problem was the absence of reflection of the population and the absurd believe of all information distributed by social media (internet, first of all). The negative effect of the lack of adequate information became a great challenge during Pandemic. It is important to stress this point.
Also, the authors should include some kinds of recommendations for the families and for the system of education in general and in particular way, for the children with ADHD. The article seams to provide general information for all children, who suffered terrible isolation during Pandemic, and not specifically for children with ADHD.
The authors should include the goals for future research related to the data of the article.
Author Response
We want to thank you for your thoughtful contributions to this paper. Our responses are provided in a word document for easier readership. However, they are also pasted below.
Reviewer 2
Comment 1: Dear authors, I read the paper with interest and I agree with a need of more qualitative research. However, I find a few aspects of the article, which might be improved.
Response to Comment 1: Thank you very much for the support. We have aimed to address your suggestions below and throughout the manuscript.
Comment 2: There is no kind of theoretical psychological background.
Response to Comment 2: We have included a description of the Dynamic Developmental Theory of ADHD and the implications it presents for this study (page 2, line 88-98). We also discuss the socio-ecological model on page 3 (line 112) which is a theory on human development and psychology, and the way it shapes the current study. Both theories point to the essential roles that intrapersonal, interpersonal, and broader external factors play in ADHD, family function and physical activity behaviour.
Comment 3: There is no references and comments about the age of the children with ADHD. Also, the mention of the age of the children is very important for understanding of organization of their activities. Outdoor activities, the guidance and orientation from the adults changes according to psychological age of the children and adolescents. This aspect should be covered and commented in the article.
Response to Comment 3: This is a good point. We have included a description of the role of children’s age in our study on page 3 (line 106), page 13 (line 538), page 16 (line 678). We have also clarified in Table 1 that the age range of child participants was 7-12 years of age (average age of 10 years old).
Comment 4: Also, all the results are based only on the responses of the questions of the interviews. A strong limitation is the absence of any kind of professional psychological or neuropsychological assessment of the population of children with ADHD. Some recent neuropsychological studies (Machinskaya and Cols.,) have shown that children with ADHD frequently show insufficient level of brain subcortical activation (brainstem system), the data, which is close to the introduction of the paper. It is important to include citations from clinical recent studies (Glozman and Cols., 2020).
Response to Comment 4: It would certainly have been informative to understand where on the ADHD spectrum the child participants in the current study were positioned. This could have provided nuance to the study interpretations, such as perhaps higher functioning children with ADHD and their caregivers were less negatively impacted by the pandemic, as they perhaps had better compensatory strategies during times of high stress. We have included this as an important area for future research on page 20 (section 4.6, line 878). We have also included information about the neuropsychological and anatomical differences in the ADHD brain vs. neurotypical brain, and how physical activity may serve to help modify brain structure and function (page 2, line 54-62).
Comment 5: It is also important to mention, that the families, actually, had opportunity to spend time outside during Pandemic. The problem was the absence of reflection of the population and the absurd believe of all information distributed by social media (internet, first of all). The negative effect of the lack of adequate information became a great challenge during Pandemic. It is important to stress this point.
Response to Comment 5: Thank you for this point. Indeed, several families indicated that they were able to participate in more physical activity as they were able to participate in more leisurely activity such as going for walks and hikes and bike rides. We have indicated on page 13 (line 546) that lack of adequate information and perhaps the fear associated with misinformation may also have contributed to a lack of physical activity engagement.
Comment 6: Also, the authors should include some kinds of recommendations for the families and for the system of education in general and in particular way, for the children with ADHD. The article seams to provide general information for all children, who suffered terrible isolation during Pandemic, and not specifically for children with ADHD.
Response to Comment 6: We have included specific recommendations for children with ADHD and their caregivers on page 19 (section 4.4, line 837) related to the important factors that facilitate physical activity in these families. We have also included information about how the education system could facilitate greater physical activity opportunities to promote physical and mental wellness for children with ADHD and their caregivers. This information is found on page 18 (line 798-802).
Comment 7: The authors should include the goals for future research related to the data of the article.
Response to Comment 7: We have included future directions on page 20 (section 4.6, line 877).

Reviewer 3 Report
Firstly, I am writing to express my gratitude for the opportunity to review the research article “Impact of COVID-19 on physical activity in families managing ADHD and the cyclical effect on worsening mental health”. I am honored to have been selected to contribute to the peer-review process for Brain Sciences.
I understand the critical importance of rigorous evaluation in academic research and am eager to lend my expertise to this process. I am confident that my analysis will be of value to the authors and help ensure that the work is of the highest quality.
In summary, this study aimed to understand how the COVID-19 pandemic affected physical activity levels in children with ADHD and their caregivers. 33 participants were interviewed, and content analysis revealed that physical activity declined for both children and caregivers due to social isolation and rising intrapersonal difficulties. These difficulties included diminishing self-efficacy, energy levels, and increased mental health problems. Participants identified a need for community support programs that offer virtual physical activity classes and psycho-emotional support groups. Supporting physical activity opportunities during high-stress situations in families managing ADHD is essential to buffer against diminishing mental wellbeing and promote cognitive, psychological, and emotional benefits.
I would like to provide a series of guidelines for each section of the article.
Introduction
The authors of the scientific article have conducted an extensive and comprehensive literature review in the introduction section, which highlights their knowledge and expertise in the field. The literature review provides a clear and concise summary of the current state of research, identifies gaps in the existing literature, and establishes the significance of the research question.
The authors have successfully synthesized the literature, providing a theoretical framework for their study, and developing a solid foundation for their research methodology. By doing so, they have demonstrated a deep understanding of the topic, and have highlighted the potential impact and contribution of their research.
Moreover, the literature review allows the reader to comprehend the importance of the research question, and to understand how the study fills a gap in the current literature. The authors have effectively used the literature review to position their research within the broader context of the field, and to highlight the relevance of their study to the scientific community.
In conclusion, the authors have conducted a rigorous and well-executed literature review, which is an essential component of any scientific article. Their literature review establishes the importance of their research question and provides a solid foundation for their study, ultimately contributing to the advancement of the field.
Materials and Methods
"In the participants section, they state: 'who was living with them for at least some of the time during the COVID-19 pandemic.' Could you please be more specific and report the average amount of time that the children and their caregivers lived together?"
In section 2.1, please add that sociodemographic data on the sample can be found in Table 1 (e.g., mean age of the children, gender distribution...)
When the authors say... “Multiple measures were implemented to ensure trustworthiness by considering credibility, dependability, and transparency”. Please explicitly state the exact measures and results of those measures. For instance, I recommend that the authors report the kappa index of agreement between the two judges who assessed the interviews with families and children.
Results
When starting the results section, it is redundant to restate the research questions. I would remove the following paragraph: “The current study explored the following three research questions: 1) How did the COVID-19 pandemic impact physical activity participation among families with children who have ADHD?; 2) What were the barriers to physical activity participation during the COVID-19 pandemic among families with children who have ADHD?; and 3) What supports do families with children who have ADHD need to better engage in physical activity during a pandemic?”
BIG PROBLEM. As a researcher, it is important to emphasize that presenting study results solely through frequencies without conducting any statistical analysis lacks the minimum methodological rigor necessary. Frequency tables are useful for describing data, but they do not provide any information on the significance or reliability of the findings. Therefore, it is essential to perform appropriate statistical analyses to determine the validity of the results and draw meaningful conclusions from the data. In conclusion, any research article that lacks statistical analysis cannot be considered scientifically rigorous.
Discussion
It is important to note that the research questions have not been adequately addressed using the methodology employed. The chosen methodology may not have been appropriate for addressing the research questions, or the study may have been underpowered or poorly designed.
It is essential that the methodology used is appropriate for addressing the research questions to produce valid and reliable results. If the research questions are not answered adequately, the research may be inconclusive, and the findings may not be reliable.
In conclusion, it is crucial to ensure that the methodology used in a scientific study is appropriate for addressing the research questions. Failure to do so can result in inconclusive findings and a lack of validity and reliability in the study.
Conclusions
As a researcher conducting a review of a scientific article, it is clear that the study has potential; however, there is a need for the authors to make an effort to conduct appropriate statistical analyses to validate their claims.
The importance of statistical analysis cannot be overstated as it allows for the identification of significant findings, the determination of the strength of the relationship between variables, and the validation of research hypotheses.
Therefore, we strongly recommend that the authors make an effort to conduct appropriate statistical analyses to support their claims and provide a valid and reliable interpretation of their results. By doing so, the study's potential impact and contribution to the field will be significantly enhanced.
In conclusion, we encourage the authors to make an effort to conduct appropriate statistical analyses and ensure that their findings are valid and reliable. By doing so, the study's potential impact will be enhanced, and the scientific community will benefit from the valuable insights gained through the research.
Best regards,
Author Response
Thank you for your comments and suggestions. Our responses are pasted below and also provided in a word document (attached).
Reviewer 3
Comment 1: Firstly, I am writing to express my gratitude for the opportunity to review the research article “Impact of COVID-19 on physical activity in families managing ADHD and the cyclical effect on worsening mental health”. I am honored to have been selected to contribute to the peer-review process for Brain Sciences. I understand the critical importance of rigorous evaluation in academic research and am eager to lend my expertise to this process. I am confident that my analysis will be of value to the authors and help ensure that the work is of the highest quality. In summary, this study aimed to understand how the COVID-19 pandemic affected physical activity levels in children with ADHD and their caregivers. 33 participants were interviewed, and content analysis revealed that physical activity declined for both children and caregivers due to social isolation and rising intrapersonal difficulties. These difficulties included diminishing self-efficacy, energy levels, and increased mental health problems. Participants identified a need for community support programs that offer virtual physical activity classes and psycho-emotional support groups. Supporting physical activity opportunities during high-stress situations in families managing ADHD is essential to buffer against diminishing mental wellbeing and promote cognitive, psychological, and emotional benefits. I would like to provide a series of guidelines for each section of the article.
Response to Comment 1: We sincerely appreciate your thoughtfulness in reviewing this work and have aimed to address your guidelines as effectively as possible.
Introduction
Comment 2: The authors of the scientific article have conducted an extensive and comprehensive literature review in the introduction section, which highlights their knowledge and expertise in the field. The literature review provides a clear and concise summary of the current state of research, identifies gaps in the existing literature, and establishes the significance of the research question. The authors have successfully synthesized the literature, providing a theoretical framework for their study, and developing a solid foundation for their research methodology. By doing so, they have demonstrated a deep understanding of the topic, and have highlighted the potential impact and contribution of their research. Moreover, the literature review allows the reader to comprehend the importance of the research question, and to understand how the study fills a gap in the current literature. The authors have effectively used the literature review to position their research within the broader context of the field, and to highlight the relevance of their study to the scientific community. In conclusion, the authors have conducted a rigorous and well-executed literature review, which is an essential component of any scientific article. Their literature review establishes the importance of their research question and provides a solid foundation for their study, ultimately contributing to the advancement of the field.
Response to Comment 2: Thank you very much for the thoroughness of your reading and your supportive comments about the introduction.
Materials and Methods
Comment 3: "In the participants section, they state: 'who was living with them for at least some of the time during the COVID-19 pandemic.' Could you please be more specific and report the average amount of time that the children and their caregivers lived together?"
Response to Comment 3: Unfortunately, we did not collect information about the specific amount of time that children and caregivers lived together. The wording “for at least some of the time” was used to be inclusive of families with diverse shared custody arrangements, but we did not inquire further about duration of co-living.
Comment 4: In section 2.1, please add that sociodemographic data on the sample can be found in Table 1 (e.g., mean age of the children, gender distribution...)
Response to Comment 4: We have included in section 2.1 that demographic information can be found in Table 1.
Comment 5: When the authors say... “Multiple measures were implemented to ensure trustworthiness by considering credibility, dependability, and transparency”. Please explicitly state the exact measures and results of those measures. For instance, I recommend that the authors report the kappa index of agreement between the two judges who assessed the interviews with families and children.
Response to Comment 5: Details regarding how trustworthiness was achieved is presented on page 5 (section 2.4.1.). Importantly, we did not include the kappa index of agreement as the two researchers worked together at all stages of coding and data analysis, which we describe in the paper as investigator triangulation (page 5, line 227). They did not complete the coding completely independently and were thus able to ongoingly rectify any inconsistencies in coding as the process unfolded.
Results
Comment 6: When starting the results section, it is redundant to restate the research questions. I would remove the following paragraph: “The current study explored the following three research questions: 1) How did the COVID-19 pandemic impact physical activity participation among families with children who have ADHD?; 2) What were the barriers to physical activity participation during the COVID-19 pandemic among families with children who have ADHD?; and 3) What supports do families with children who have ADHD need to better engage in physical activity during a pandemic?”
Response to Comment 6: We have removed the paragraph restating the research questions.
Comment 7: BIG PROBLEM. As a researcher, it is important to emphasize that presenting study results solely through frequencies without conducting any statistical analysis lacks the minimum methodological rigor necessary. Frequency tables are useful for describing data, but they do not provide any information on the significance or reliability of the findings. Therefore, it is essential to perform appropriate statistical analyses to determine the validity of the results and draw meaningful conclusions from the data. In conclusion, any research article that lacks statistical analysis cannot be considered scientifically rigorous.
Response to Comment 7: While some qualitative research can include inferential statistics such as Chi-square analyses to determine differences between groups or categories, our research questions did not have a comparative focus. In other words, we were not empirically evaluating whether one theme was statistically more significant than another theme. This level of dissection and categorization was not appropriate for our research goals which were to characterize the experiences of children with ADHD and their caregivers, not to hierarchically organize their relevance. There is a rich history of qualitative research using non-inferential statistical techniques such as grounded theory, thematic analysis, and content analysis (which is what we used in the current study). These techniques focus on identifying patterns, themes and relationships in the data and drawing conclusions based on those patterns. They do not require inferential statistics to be considered rigorous, valid, or reliable. In qualitative analysis it is essential to maintain rigor and transparency in the research process by defining the research question, developing a systematic data collection and analysis process, and providing detailed documentation of the findings. It is equally important to ensure that the data are collected and analyzed in an unbiased manner. We have indicated on page 5 how our data analysis methodology meets all these standards and have provided references for how this aligns with the literature’s standards for rigorous qualitative science.
Archibald, M. M. Investigator triangulation: A collaborative strategy with potential for mixed methods research. J. Mixed Methods Res. 2016, 10, 228-250. doi: 10.1177/1558689815570092
Flick, U. Triangulation in qualitative research. A companion to qualitative research, 2004, 3, pp. 178-183.
Patton MQ. Enhancing the quality and credibility of qualitative analysis. Health services research. 1999 Dec;34(5 Pt 2):1189. Available from: https://www.ncbi.nlm.nih.gov/pmc/articles/PMC1089059/.
Discussion
Comment 8: It is important to note that the research questions have not been adequately addressed using the methodology employed. The chosen methodology may not have been appropriate for addressing the research questions, or the study may have been underpowered or poorly designed. It is essential that the methodology used is appropriate for addressing the research questions to produce valid and reliable results. If the research questions are not answered adequately, the research may be inconclusive, and the findings may not be reliable. In conclusion, it is crucial to ensure that the methodology used in a scientific study is appropriate for addressing the research questions. Failure to do so can result in inconclusive findings and a lack of validity and reliability in the study.
Response to Comment 8: Please see our response to Comment 7.
Conclusions
Comment 9: As a researcher conducting a review of a scientific article, it is clear that the study has potential; however, there is a need for the authors to make an effort to conduct appropriate statistical analyses to validate their claims. The importance of statistical analysis cannot be overstated as it allows for the identification of significant findings, the determination of the strength of the relationship between variables, and the validation of research hypotheses. Therefore, we strongly recommend that the authors make an effort to conduct appropriate statistical analyses to support their claims and provide a valid and reliable interpretation of their results. By doing so, the study's potential impact and contribution to the field will be significantly enhanced. In conclusion, we encourage the authors to make an effort to conduct appropriate statistical analyses and ensure that their findings are valid and reliable. By doing so, the study's potential impact will be enhanced, and the scientific community will benefit from the valuable insights gained through the research.
Response to Comment 9: We appreciate the reviewer’s conviction for statistical rigor and inferential analysis. However, we disagree that all forms of rigorous science must involve comparative inferential statistical analysis. Please see our response to Comment 7 above for more details.

Round 2
Reviewer 2 Report
I have read the comments of the authors and I find them correct and pertinent.
Reviewer 3 Report
Dear editor,
I have read the authors' response to my revisions, and I believe that I am not adequately qualified to evaluate the methodology of this article. Therefore, I understand that I am not in a position to pass judgment on its acceptance or rejection, and I would prefer to withdraw as a reviewer. Thank you very much, and I apologize for any inconvenience.
Regards,